# Integrase-Defective Lentiviral Vectors for Delivery of Monoclonal Antibodies against Influenza

**DOI:** 10.3390/v12121460

**Published:** 2020-12-17

**Authors:** Zuleika Michelini, Judith M. Minkoff, Jianjun Yang, Donatella Negri, Andrea Cara, Brendon J. Hanson, Mirella Salvatore

**Affiliations:** 1Department of Medicine, Weill Cornell Medical College, New York, NY 10021, USA; zuleika.michelini@iss.it (Z.M.); judyfontanaminkoff@gmail.com (J.M.M.); jiy2017@med.cornell.edu (J.Y.); 2Department of Infectious Diseases, Istituto Superiore di Sanità, 00161 Rome, Italy; donatella.negri@iss.it; 3National Center for Global Health, Istituto Superiore di Sanità, 00161 Rome, Italy; acara@iss.it; 4Defence Medical and Environment Research Institute, DSO National Laboratories, Singapore 117510, Singapore; hbrendon@dso.org.sg

**Keywords:** integrase-defective lentiviral vector, influenza, monoclonal antibody, passive immunity, genetic immunization

## Abstract

Delivering rapid protection against infectious agents to non-immune populations is a formidable public health challenge. Although passive immunotherapy is a fast and effective method of protection, large-scale production and administration of monoclonal antibodies (mAbs) is expensive and unpractical. Viral vector-mediated delivery of mAbs offers an attractive alternative to their direct injection. Integrase-defective lentiviral vectors (IDLV) are advantageous for this purpose due to the absence of pre-existing anti-vector immunity and the safety features of non-integration and non-replication. We engineered IDLV to produce the humanized mAb VN04-2 (IDLV-VN04-2), which is broadly neutralizing against H5 influenza A virus (IAV), and tested the vectors’ ability to produce antibodies and protect from IAV in vivo. We found that IDLV-transduced cells produced functional VN04-2 mAbs in a time- and dose-dependent fashion. These mAbs specifically bind the hemagglutinin (HA), but not the nucleoprotein (NP) of IAV. VN04-2 mAbs were detected in the serum of mice at different times after intranasal (i.n.) or intramuscular (i.m.) administration of IDLV-VN04-2. Administration of IDLV-VN04-2 by the i.n. route provided rapid protection against lethal IAV challenge, although the protection did not persist at later time points. Our data suggest that administration of mAb-expressing IDLV may represent an effective strategy for rapid protection against infectious diseases.

## 1. Introduction

Several animal studies have highlighted the importance of neutralizing monoclonal antibodies (mAbs), or mixtures of mAbs, for the prevention and control of the most severe global infectious diseases [1]. Nevertheless, mAbs are expensive and require multiple administrations to be effective. Therefore, more cost-effective delivery methods are needed. Genetic immunization by viral vector-mediated delivery of mAbs has been recently proposed as a better alternative to the direct injection of mAbs for both short- and long-term therapy [2,3,4,5,6].

Among the currently available viral vectors, integrase-defective lentiviral vectors (IDLV) are a safe and effective delivery system with the possible advantage of early and sustained transgene production and the safety feature of non-integration of the viral nucleic acid into the host genome. In fact, IDLV have a mutation in their integrase coding sequence that blocks DNA integration, thereby drastically reducing the risk of unintentional insertion of the vector into a critical host gene during treatment, while still maintaining high transduction efficiency [7]. There is strong evidence that the relative integration frequencies of replication-defective IDLV, such as the vectors used in this study, are 3–4 log reduced, as compared to those obtained using conventional integration-competent lentiviral vectors in both cell culture systems and in vivo [8,9,10,11,12,13,14,15,16,17,18]. This recombination frequency is within the range described when using plasmid DNA in vitro [19,20] or in vivo [21,22], and below the frequency of spontaneous gene-inactivating mutations, using a worse case analysis (see [23] for a discussion of this calculation). As a consequence, the genotoxicity of IDLV-associated insertional mutagenesis/oncogenesis is negligible, thus making IDLV highly attractive from a biosafety standpoint.

IDLV are also replication-deficient, since all structural proteins required to construct the vector are supplied *in trans* to the packaging signal [24], and self-inactivating due to a deletion in the 3′ long terminal repeat region of the viral promoter and enhancer sequences [25], so that they complete only a single round of infection. In addition, the fact that viral genes are only expressed from plasmids during vector production means that they are unaffected by the low fidelity of reverse transcriptase, further minimizing the possibility of reversion of the integrase mutation. IDLV are routinely pseudotyped with the surface protein G of VSV (VSV-G), which confers high transduction efficiency and broad host cell range [7,26,27,28]. Finally, protein expression from IDLV is stable and persistent in non-dividing cells [7,16,29,30,31,32,33,34] We have previously shown that IDLV engineered to express influenza A virus (IAV) antigens can elicit protective immunity in vivo [35].

In the present study, we developed IDLV that produce mAbs that are protective against IAV. The mAb we chose to express from IDLV, called VN04-2, is a well-studied mAb that targets the hemagglutinin (HA) surface protein of highly pathogenic H5N1 IAV, and it has proven prophylactic activity in vivo [36]. We showed that both intranasal (i.n.) and intramuscular (i.m.) administration of IDLV are able to stimulate the production of specific and functional mAbs that protect against H5 IAV in a mouse model.

## 2. Material and Methods

### 2.1. IDLV Production

IDLV producing VN04-2 mAbs or expressing green fluorescent protein (GFP) or nucleoprotein (NP) were produced by three-plasmid co-transfection in LentiX 293T cells (Clontech, Mountain View, CA, USA) as previously described [35]. Plasmids included: 1. the pTY2-CMV transfer plasmid; 2. the pCHelp/IN- packaging plasmid; and 3. the Env pMD.G plasmid expressing VSV-G [35]. For the transfection, cells were plated on 100 mm tissue culture-treated dishes (Corning, Tewksbury, MA) coated with 0.8% gelatin (Millipore-Sigma, Burlington, MA, USA) and 0.002% poly L-lysine (Millipore-Sigma, Burlington, MA, USA), and incubated overnight. Cells were transfected with the three plasmids described above at a ratio 8:8:4 using the Profection Mammalian Transfection System (Promega Corporation, Madison, WI, USA), and media were changed after 10 h. At 48 and 72 h post-transfection, culture supernatants were collected, cleared of cellular debris by centrifugation, filtered through a 0.45 µm pore-sized polyvinylidene difluoride (PVDF) filter (EMD Millipore, Billerica, MA, USA), and concentrated on a 20% sucrose gradient (Sigma-Aldrich) by ultracentrifugation at 27,000 rpm (~131,100× *g*) for 2 h at 4 °C using an AH-629 swinging bucket rotor (ThermoFisher Scientific, Waltham, MA, USA) in a Sorvall WX Ultra 80 ultracentrifuge (ThermoFisher Scientific, Waltham, MA, USA). Pellets were resuspended in PBS and stored at −80 °C until use [35]. Levels were normalized by quantitative determination of retroviral reverse transcriptase (RT) activity using the Reverse Transcriptase Assay, Colorimetric (Roche, Indianapolis, IN, USA) or by HIV-1 p24 ELISA assays (PerkinElmer, Waltham, MA, USA).

### 2.2. Western Blot

293T LentiX cells were transduced with increasing amounts of IDLV-VN04-2. Equal volumes of supernatants collected post-transduction were analyzed by sodium dodecyl sulfate polyacrylamide gel electrophoresis (SDS–PAGE) using 4–15% polyacrylamide gels (Bio-Rad Laboratories, Inc., Hercules, CA, USA) and probed by Western blot using peroxidase-labeled anti-human IgG (H + L) (KPL, Gaithersburg, MD, USA) to detect the heavy and light chains. Protein bands were visualized by Gelcode Blue Stain reagent (ThermoFisher Scientific, Waltham, MA, USA). For detecting the production of mAbs in mouse serum, samples (200 µg of total protein, 1–2 µL of serum) were probed with rabbit anti-human IgG antibody (Rabmab, Abcam Cambridge, MA, USA) and goat anti-rabbit poly-HRP (Thermo Scientific Pierce, Waltham, MA, USA).

### 2.3. Dot Blots

Recombinant H5 HA from influenza A/Vietnam/1203/04 Biodefense and Emerging Infections Research Resources Repository (BEI Resources, NIAID/NIH, Manassas, VA, USA) (2 μg) and recombinant influenza NP (Imgenex Corporation, San Diego, CA, USA) (5 μg), as a specificity control, were resuspended in PBS, spotted on nitrocellulose and allowed to dry. The blots were then blocked for 2 h at room temperature in PBS with 5% milk plus 0.05% Tween20, washed 3 times with PBS plus 0.2% Tween 20 and probed with 1:200 dilutions of supernatant from cells transduced with IDLV-VN04-2 or from untransduced cells. A 1:200 dilution of VN04-2 (BEI Resources, NIAID/NIH, Manassas, VA, USA) was used as a positive control for H5 HA detection. A 1:5000 dilution of peroxidase-labeled anti-human IgG (H + L) was used to detect bound antibodies.

### 2.4. Influenza Viruses

Influenza A/Vietnam/1203/2004(H5N1)-PR8-IBCDC-RG/GLP (VNH5N1-PR8 IAV), a reassortant virus strain which contains HA and NA of IAV Vietnam/1203/2004 (H5N1, phylogenetic clade 1) and six internal gene segments of A/Puerto Rico/8/34 (H1N1) was obtained from the CDC International Reagent Resources (IRR, Manassas, VA, USA). The mouse-adapted influenza H1N1 pdm2009 and A/Philippines/2/1982 (IAV Phil H3N2 phylogenetic clade 2) viruses were a gift from Dr. Palese, Mount Sinai School of Medicine, NY.

### 2.5. Mouse Immunizations and Challenge Experiments

Female BALB/c mice, 6–8 weeks old, were purchased from Charles River NCI or Jackson Laboratories. Mice were anesthetized with inhalational isoflurane and each mouse received 160–500 ng RT of IDLV-VN04-2, or IDLV expressing other proteins, as noted in the text (IDLV-NP, IDLV-GFP), via either the i.m. (in 100 µL PBS) or i.n. route (in 50 µL PBS).

Mice were challenged at different times after immunization with 2–5 LD50 of IAV in 50 µL of PBS and monitored daily for weight loss and survival. Mice with a loss of ≥20% of their initial body weight were euthanized. Serum samples were collected by retro-orbital blood collection and stored at −20 °C until analyzed. All animal procedures were performed in accordance with Institutional Animal Care and Use Committee (IACUC) guidelines and have been approved by the IACUC of the Joan & Sanford I. Weill Medical College of Cornell University (Protocol Number: 2009–045). All procedures were performed under inhalational isoflurane anesthesia, and every effort was made to minimize suffering.

### 2.6. H5 Enzyme-Linked Immunosorbent Assay (ELISA)

Levels of serum VN04-2 mAbs were determined using a modified indirect ELISA. Briefly, a 96-well plate (high binding microplate, Greiner Bio-One, Rainbach, Austria) was coated with 0.5 μg/well of recombinant HA protein from IAV Vietnam/1203/2004 (H5N1), (Protein Sciences Corp. Meriden, CT, USA, Lot No 1368-142). Plates were blocked with 200 μL of blocking buffer (PBS 1X and 2% milk (Bio-Rad, Hercules, CA, USA)). After each incubation step, plates were washed four times with PBS-0.01% Tween20. Samples were diluted 1:10 in blocking buffer, added to the plates, and incubated for two hours at room temperature. After washing, wells were incubated with 1:1000 of affinity-purified goat anti-human IgG Fc horseradish peroxidase (HRP) conjugate (Novex by Life technologies, Carlsbad, CA, USA) for one hour, then washed again. TMB Substrate Solution (ThermoFisher Scientific, Waltham, MA, USA) was added and the reaction was stopped with 2N H2SO4 (ThermoFisher Scientific, Waltham, MA, USA). The results were read and recorded at optical density (O.D.) 450 nm using Magellan software and the Sunrise Absorbance Reader (Tecan, Mannedorf, Switzerland). A standard curve for the assay (ranging from 0.11 to 7.5 ng/mL of IgG) was obtained from IgG released in the supernatants of 293T Lenti-X cells at 72 h after transduction with IDLV-VN04-2, and quantified using the Easy-Titer™ Human IgG (H + L) Assay Kit (ThermoFisher Scientific, Waltham, MA, USA). The Easy-Titer Human IgG (H + L) Kit was also used to quantify the total amount of IgG in the cell supernatants.

### 2.7. Quantification of IAV RNA

The lungs of mice were homogenized in PBS and viral RNA was extracted using the QIAamp viral RNA mini kit (Qiagen, Germantown, MD, USA). Viral RNA was then quantified by one-step real-time RT-PCR assay (qPCR). Primers and probes were obtained from BEI Resources (ATCC, Manassas, VA, USA NR15593, 15594 and 15595). For quantification, a standard curve was obtained using viral RNA that was extracted from IAV-infected cell supernatants that were previously titered by plaque assay. Briefly, 5 μL aliquots of RNA extracted from the lungs of mice were mixed with 20 µL of premixed reaction solution (SuperScript III Platinum One-Step qRT-PCR Kits, (ThermoFisher Scientific, Waltham, MA, USA) containing 40 µM each of the forward and reverse primers, and 10 µM of the probe. Amplification and detection were performed using the StepOnePlus Real-Time PCR System (Applied Biosystem, Foster City, CA, USA). Amplification conditions were 30 min at 50 °C, 2 min at 95 °C, and 45 cycles of 15 s at 95 °C and 30 s at 55 °C. Samples were run in duplicate, the standard curve was performed in triplicate, and ribonuclease-free water served as the negative control. Nucleic acid extraction, validation, and normalization were performed with TaqMan Rodent GAPDH Control Reagents (Applied Biosystem, Foster City, CA, USA).

## 3. Results

### 3.1. Generation of IDLV Expressing Anti-IAV HA mAbs

To assess whether IDLV could be an effective antibody-delivery platform, we chose to express from IDLV, as proof of concept, the mAb VN04-2, a well-studied mAb with proven in vivo efficacy. This mAb was raised against the HA of influenza A/Vietnam/1203/04, and is broadly neutralizing against highly pathogenic H5N1 influenza viruses in a clade-independent fashion [36,37]. VN04-2 mAbs have been shown to protect mice from lethal challenge when administered in low doses (1 μg/kg body weight) [36]. Analysis of VN04-2 escape variants of IAV indicated that at least three mutations within the HA antigenic region are required before escape is possible, making escape more unlikely [38]. To generate the transfer plasmid which would allow expression of VN04-2 from the IDLV vector, we inserted the huG1-VN04-2 mAb expression cassette from pCMV-huG1-VN04-2, a plasmid which codes for humanized antibodies containing constant regions from human IgG1 and variable regions from mouse VN04-2 [36], into our transfer vector [35] to obtain pTY2-CMV-huG1-VN04-2. We also created an additional transfer vector, pCMV-huG1, which codes for humanized antibodies containing constant regions from human IgG1, but lacking the VN04-2 variable regions for both heavy and light chains [36]. These vectors were then used in triple co-transfection experiments to create IDLV expressing humanized VN04-2 mAbs (IDLV-VN04-2) and IDLV-HuG1 expressing a mAb that lacks the VN04-2 variable regions for both heavy and light chains, as a control. The plasmids used in this system included (i) the pTY2-CMV-huG1-VN04-2 transfer plasmid, which expresses VN04-2 [39] (ii) the integrase-defective packaging plasmid, pCHelp/IN- [35], which contains all necessary packaging elements as well as a point mutation (D116N) that inactivates the function of the integrase protein and (iii) the Env plasmid, which expresses VSV-G that is used to pseudotype the resulting vector (Figure 1).

### 3.2. Cells Transduced with IDLV-VN04-2 Produce HA-Binding mAbs in a Time- and Dose-Dependent Fashion

To evaluate the production of VN04-2 mAbs in vitro, 293T LentiX cells were transduced with increasing amounts of IDLV-VN04-2. Cell supernatants were harvested at 24 (not shown), 48 and 72 h post-transduction, and the presence of VN04-2 mAbs was assayed by Western blot. The results showed that VN04-2 mAbs were produced, and that they accumulated in a clear time- and dose-dependent manner (Figure 2a,b). To confirm this finding, we also measured the levels of IgG in the supernatants. We found that cells transduced with 500 ng p24 of IDLV-VN04-2 produced levels of IgG equivalent to 260 ng/mL at 48 h and 769 ng/mL at 72 h. Levels of IgG detected from cells transduced with 1 μg p24 of IDLV-VN04-2 were 392 ng/mL at 48 h and 829 ng/mL at 72 h. Untransduced samples and samples obtained from cells transduced with 50 or 125 ng p24 IDLV-VN04-2 contained IgG levels that were below the limit of detection (<15.6 ng/mL).

To demonstrate that mAbs produced by IDLV-VN04-2 were functional and specific to H5 HA, we performed dot-blot experiments. Recombinant H5 HA from influenza A/Vietnam/ 1203/04 and recombinant IAV NP as a specificity control were spotted onto nitrocellulose. The nitrocellulose was then dried and probed with supernatant derived from cells transduced with IDLV-VN04-2, or from untransduced cells as controls. Figure 3 shows that the supernatant from transduced cells, but not from untransduced cells, specifically bound to undenatured H5 HA, suggesting that VN04-2 mAbs produced by IDLV are functional.

### 3.3. A Single Administration of IDLV-VN04-2 Induces the Production of mAbs In Vivo

To evaluate whether IDLV could produce mAbs *in vivo,* we administered IDLV-VN04-2 to groups of 3 BALB/c mice, either by the i.n. or i.m. route. Mice were bled before and at 28 days after administration. VN04-2 mAbs from mouse serum were visualized by Western blot probed with anti-human IgG. As shown in Figure 4a, a distinct band corresponding to the size of the heavy chain of the VN04-2 antibody (~50 kDa) was detected at day 28 after administration by the i.n. or i.m. route.

In a separate experiment, groups of mice received IDLV-VN04-2 by the i.n. route. Unimmunized mice were used as controls. The presence of humanized VN04-2 mAbs in the serum of these mice was measured by ELISA at various time points up to 28 days after IDLV administration. The results shown in Figure 4c,d indicate that production of VN04-2 mAbs peaked around 2–7 days after administration and was detected through the end of the experiment. No detectable anti-H5 HA antibodies were found in serum from naïve control mice (not shown). To further investigate if the production of mAbs in vivo could be affected by the genetic background or immune status of the mice, we repeated the experiment in C57BL/6 and interferon (IFN) α/β receptor knock-out mice. Administration of IDLV-VN04-2 by the i.n. route was able to induce the production of mAbs in both of these mouse strains (Appendix A).

### 3.4. IDLV-VN04-2 Protects Mice From IAV Challenge

We next assessed if mAbs produced by IDLV-VN04-2 are able to protect mice from IAV challenge. To this end, groups of BALB/c mice were inoculated with IDLV-VN04-2 by the i.n. or i.m. route. Untreated mice were used as controls. Three days after administration, mice were challenged with 5 LD50 of VNH5N1-PR8 IAV. Mice were monitored for weight loss (Figure 5a) and survival (Figure 5b). All mice that received IDLV-VN04-2 by the i.n. route and 40% of the mice that received IDLV-VN04-2 by the i.m. route were protected from IAV challenge. In contrast, all untreated mice succumbed to IAV infection. Moreover, real-time PCR analysis of RNA extracted from lungs showed a decreased number of influenza viral RNA copies in mice immunized by the i.n. route (Figure 5c). Given the improved protection achieved by i.n. immunization, this route was chosen for the following studies.

To ensure that protection was due to the production of VN04-2 mAbs, and not to a non-specific effect of the vector, we repeated the experiment using either IDLV-VN04-2 or an IDLV expressing green fluorescent protein (IDLV-GFP). In this set of experiments, we administered IDLV-VN04-2 or IDLV-GFP by the i.n. route to groups of BALB/c mice. Untreated mice were used as controls. The mice were then lethally challenged with VNH5N1-PR8 IAV at 4 days after IDLV administration. The results shown in Figure 5d,e show that mice that received IDLV-VN04-2 were protected from challenge while mice that received IDLV-GFP and naïve mice lost weight and succumbed to the infection. The mice receiving IDLV-GFP also had higher viral RNA copy numbers in the lung than mice receiving IDLV-VN04-2 (Figure 5f).

To establish the duration of protection mediated by VN04-2 mAbs produced in vivo, in a separate series of experiments, we administered IDLV-VN04-2 by the i.n. route to groups of BALB/c mice, or we left mice unimmunized. The mice were then lethally challenged with VNH5N1-PR8 IAV at different time-points (10, 21 and 30 days) after IDLV administration. Mice that received IDLV-VN04-2 at 10 (not shown) or 21 days before challenge did not experience significant weight loss (Appendix A). These mice were also protected from lethal IAV challenge (Appendix A), while controls succumbed. Mice challenged at 30 days after administration of IDLV-VN04-2 exhibited delayed weight loss with no protection, suggesting that either the vector is completely or partially lost and/or the mAbs produced from the vector were below the level needed to prevent disease (Appendix A).

In our previous studies, we found that vaccination with IDLV expressing the conserved IAV nucleoprotein (IDLV-NP) can induce broad protective immunity against challenge with several IAV subtypes [35,40]. However, IDLV-NP requires two immunizations to be effective, thereby making it not suitable for inducing rapid protection. In the experiments described above, IDLV-VN04-2 achieved rapid protection against IAV in mice, although the protection did not persist at 30 days post-administration. We therefore assessed if co-administration of two IDLV to provide protection at later time points is possible. Groups of BALB/c mice were given either IDLV-VN04-2 or IDLV-NP alone, or IDLV-VN04-2 and IDLV-NP together or left unimmunized. Groups of mice that received IDLV-NP were given a second i.n. dose of the same vector at 2 weeks after the first IDLV administration. All mice were then challenged with influenza A/Philippines/2/1982 (IAV PhilH3N2 30 days after the initial immunization and monitored for weight loss and survival. Mice receiving a combination of IDLV-VN04-2 and IDLV-NP, but not IDLV-VN04-2 alone, were protected when challenged with IAV Phil H3N2 (Appendix A), suggesting that it is feasible to administer a combination of mAb-producing IDLV with antigen-expressing IDLV to potentially extend the protection induced by mAb-expressing IDLV.

## 4. Discussion

Several viral delivery systems have been investigated for their potential to protect against infectious agents, either through genetic immunization or by delivery of mAbs [41]. In this study, we showed that IDLV are an effective platform for producing mAbs in vivo that provide protection from lethal IAV challenge.

IDLV were more effective at protecting from IAV challenge when administered by the i.n. route, compared to i.m. administration. This route of immunization, which corresponds to the point of entry of IAV, has not been widely explored, but it may in fact be an ideal route for IDLV-based platforms, given that IDLV effectively transduce and persist in non- replicating cells, which make up almost 95% of the epithelial cell population in the airway [42]. In this respect, i.n. vector delivery could facilitate local production of neutralizing antibodies in respiratory mucosal cells, which could result in more rapid protection against infectious agents that use the respiratory route of entry, including IAV and emerging viruses like SARS-CoV-2. In our study, it is possible that the difference we observed in the effectiveness of the i.n. and i.m. routes of IDLV administration could be explained by different levels of mAbs being produced or delivered to the local respiratory environment. Another possibility is that different cell types transduced by IDLV may support higher levels of transcription or longer persistence of the vector. This is an important point that need to be investigated; however, our initial study was not designed for in-depth identification of the production of antibody at the single-cell level. Although we detected similar levels of mAbs in the serum of mice after i.n. and i.m. administration of IDLV, future studies are needed to determine the specific levels of mAbs reaching or being produced in the respiratory tract.

Since local immune responses to a viral infection can be altered by a previous infection, another possibility for these observed differences is that i.n. administration of IDLV may promote some level of nasal priming that differentially influences the response to a subsequent IAV challenge [43,44,45]. This type of priming effect has been described for other viral vectors that were administered by the i.n. route [45] and could explain the small survival advantage we observed in mice given IDLV-GFP (Figure 5c,d). It could also explain the finding that IDLV-VN04-2 alone provided some protection against H3N2 IAV, even though VN04-2 mAbs are specific for H5 HA (Appendix A).

The protection conferred by IDLV-VN04-2 in our study was most effective in the days immediately following i.n. administration and similarly to the adenovirus-based viral vector antibody delivery systems [41,45], the protection tended to wane with time. This effect is possibly due to a decrease over time in the amount of IDLV in the respiratory airways and/or the level of mAbs being produced. Future studies aimed at measuring the levels of antibody produced in the airways may be important in establishing the amount of antibody needed for protection. Recent reports have indicated that IDLV persist at steady levels and in the absence of integration for several months in a number of anatomical sites with non-replicating or slowly replicating cells, including the muscle and the respiratory tract [17,34,46,47]. In this study, we did not measure the potential for background integration of IDLV. Moreover, we did not investigate the long-term duration of the protective effects mediated by IDLV-VN04-2 after i.m. administration; however, IDLV delivered by this route did not provide the same level of protection as the i.n. route at early time points. Recent studies in antibody development have shown that certain amino acid modifications can extend the serum half-life of mAbs [48]. It is possible that optimization strategies to improve the half-lives of mAbs expressed by IDLV could improve and prolong the duration of protection elicited by our approach.

Cell-mediated immunity directed towards conserved IAV proteins, such as NP, was a key factor in determining the severity of disease during the 2009 H1N1 influenza pandemic in humans [49]. We found previously that i.n. administration of IDLV expressing IAV NP (IDLV-NP) can induce broad protective immunity against challenge with several IAV subtypes. However, this approach requires two administrations over the course of a month to be effective [35,40], thereby making it not suitable for inducing rapid protection. In the experiments described above, IDLV-VN04-2 achieved rapid protection against IAV in mice, although the protection did not persist at 30 days post-administration. Since VN04-2 mAbs produced by IDLV elicit early protection and effective NP-directed host immune responses take longer to develop, we propose that combining these IDLV could provide both rapid and long-term protection from IAV infection. In a preliminary study, we showed that mice receiving a combination of IDLV-VN04-2 and IDLV-NP, but not IDLV-VN04-2 alone, were protected from IAV challenge. This result suggests that the addition of IDLV-VN04-2 does not negatively affect the protection elicited by IDLV-NP (Appendix A). This promising result suggests that it may be possible to combine at least two different IDLV-mediated strategies to achieve broader protection from IAV. Whereas anti-HA mAbs may protect early after IDLV administration but are only partially effective longer term, adding IDLV that expresses an IAV antigen that generates more lasting immunity could provide additional protection after the immediate effect of the mAbs has waned. In particular, expressing a conserved IAV protein, like NP, that stimulates cross-protective cell-mediated immunity could also protect against multiple different strains of IAV. Future studies will aim to identify the optimal combination of IDLV that would limit the scope of infection and possibly reduce the likelihood of severe disease.

In summary, we demonstrated that IDLV can be engineered to produce functional mAbs in vitro and in vivo. In our hands, the i.n. route of IDLV administration is more effective than the i.m. route in terms of inducing early protection against IAV. This protection waned in the month after IDLV administration, possibly due to a decrease in the level of vector and/or mAbs produced over time. Optimization of the mAb sequence and/or the inclusion of additional IDLV expressing IAV antigens for vaccination could improve the potential use of IDLV as an antibody delivery system.

## Figures and Tables

**Figure 1 viruses-12-01460-f001:**
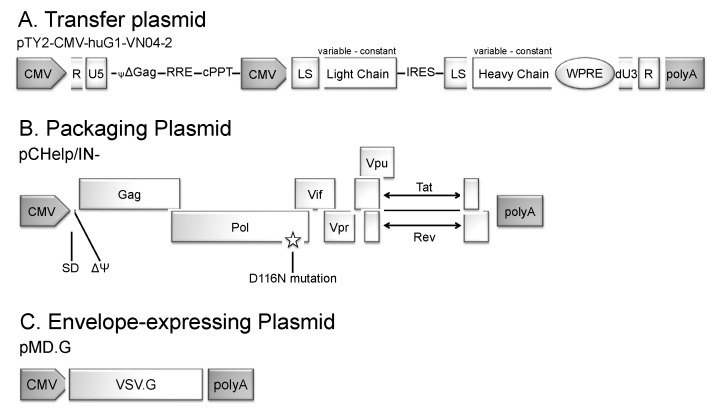
**Schematic representation of lentiviral plasmids.** (**A**). The lentiviral transfer plasmid pTY2-CMV-huG1-VN04-2 expresses VN04-2 mAbs directed against the H5 hemagglutinin (HA) of influenza A virus (IAV); (**B**). The packaging plasmid contains all necessary packaging elements and the D116N integrase-inactivating mutation; (**C**). The plasmid expressing the envelope protein of VSV-G for pseudotyping.

**Figure 2 viruses-12-01460-f002:**
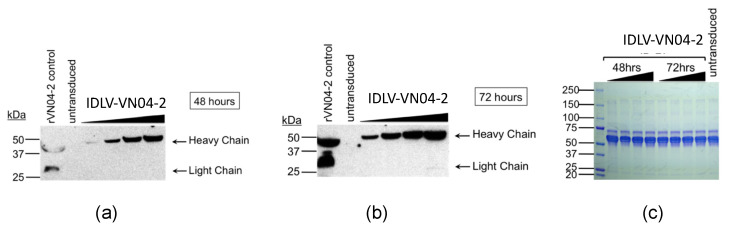
**Time- and dose-dependent production of VN04-2 mAbs.** Supernatants from 293T cells transduced with increasing doses of IDLV-VN04-2 (corresponding to 50 ng, 125 ng, 500 ng or 1 μg HIV-1 p24) or left untransduced, as a negative control, were collected at (**a**) 48 h or (**b**) 72 h post-transduction. Samples were separated on 4–15% SDS–PAGE gels and probed for human IgG by Western blot to determine VN04-2 production. Recombinant VN04-2 (rVN04-2) antibodies were run as a positive control for mAb detection. (**c**) Aliquots of the same samples depicted in panels (**a**) and (**b**) were run on a separate 4–15% SDS–PAGE gel and stained with Coomassie Brilliant Blue to visualize total protein.

**Figure 3 viruses-12-01460-f003:**
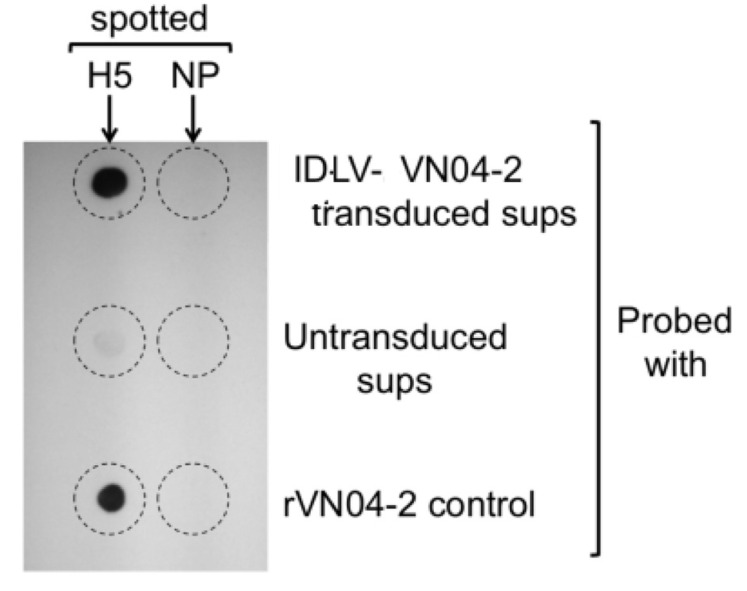
**VN04-2 mAbs generated from integrase-defective lentiviral vectors (IDLV) bind H5 HA.** Supernatants (sups) from cells transduced with IDLV-VN04-2 were used to probe dried nitrocellulose dot plots spotted with recombinant IAV H5 HA protein, or NP as specificity control. Recombinant VN04-2 mAbs (rVN04-2) were used as a positive control for detection of H5 HA. Supernatants from untransduced cells were used as a negative control.

**Figure 4 viruses-12-01460-f004:**
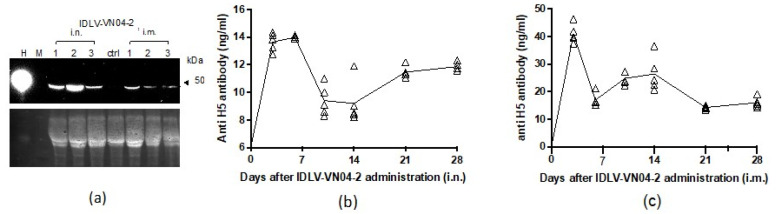
**Temporal production of VN04-2 mAbs after IDLV-VN04-2 administration in vivo.** (**a**) Presence of VN04-2 mAbs in the serum of individual mice (*n* = 3) at 28 days after receiving IDLV-VN04-2 by the intranasal (i.n.) or intramuscular (i.m.) route (200 and 500 RT units, respectively) was measured by Western blot for human IgG. Mixed human sera (H, positive control), serum from an untreated mouse (ctrl, negative control), protein standard marker (M). All in all, 100 µg of total protein was loaded for each sample. Total unstained proteins (bottom) as loading control were visualized using the ChemiDoc MP system (Bio-Rad, Hercules, CA, USA). Groups of 5 mice received (**b**) i.n. or (**c**) i.m. administration of IDLV-VN04-2 (250 and 500 RT units, respectively). Levels of serum anti-H5 antibodies before or at 3, 6, 9, 14, 21 and 28 days after IDLV-VN04-2 administration were measured by ELISA. Triangles represent individual mice; lines indicate mean values.

**Figure 5 viruses-12-01460-f005:**
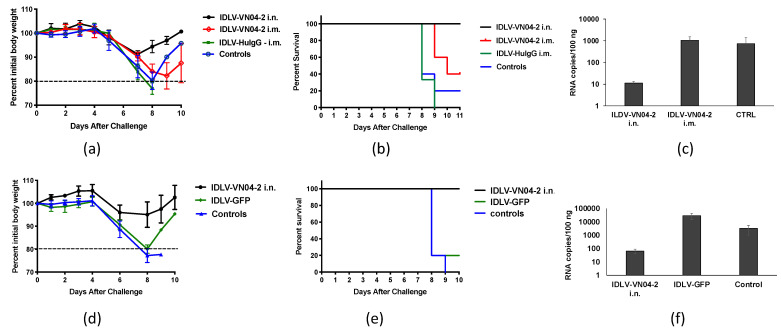
**Administration of IDLV-VN04-2 protects from H5 IAV challenge** BALB/c mice (*n* = 5) received IDLV-VN04-2 either by the i.n. or i.m. route (250 or 500 ng RT units, respectively), and IDLV-huG1 by the i.m. route or were left unimmunized as a control (**a**–**c**). In a separate experiment, mice received 250 ng RT units of either IDLV-VN04-2 or IDLV-GFP by the i.n. route or were left unimmunized as a control (**d**–**f**). Mice were lethally challenged with 2 LD50 of VNH5N1-PR8 IAV 3 days after IDLV administration and monitored for weight loss (**a**,**d**) and survival (**b**,**e**). Mice that lost ≥20% of their initial body weight were euthanized and counted as dead. Comparison of survival curve (**b**) *p* = 0.0096 and (**c**) *p* = 0.0004 Log-rank Mantel–Cox test. Protection experiments are representative of three independent replicates. Influenza copy numbers in the lungs were measured by real-time PCR and are shown in (**c**,**f**).

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
