# Peer review of "Integrase-Defective Lentiviral Vectors for Delivery of Monoclonal Antibodies against Influenza"

_viruses, 2020, doi:10.3390/v12121460_

Round 1
Reviewer 1 Report
The article Michelin Zi et al. describes the use of integrase-defective lentiviral vectors (IDLV) as a delivery system for monoclonal antibodies (mAbs) against the Influenza virus haemagglutinin. The subject could be of interest to the readers as similar approaches have recently been described. For example, recently, Tan TK et al. (Thorax 2020). have described a similar strategy using lentiviral vectors.
The advantage of the Michelin Zi et al. study is that IDLV have a higher safety profile and IDLV are more appropriate if the objective is an immunoprophylaxis approach. However, the IDLV system needs further improvement in its safety profile (such as reduce background integration) before being considered for the immunoprophylaxis approach, such as the one here described. There are some points that the authors should discuss further in the article to better highlight that this study shows a proof of principle approach.
Major concerns
- In the "IDLV Production" section (Material and Methods), the authors cite a previous publication to describe the IDLV production method used. However, I compared the article cited here and a recent article of the authors (also mentioned in other sections), and I saw that they use different transfection reagents in the most recent work. Furthermore, a packaging construct with a wildtype integrase sequence is used in the article cited. The authors should include details such as the transfection reagent used and the plasmid ratio used to provide better instructions to readers interested in repeating the study.
- It is not clear to me if pTY2-CMV-huG1-VN04-2 encodes a second generation or third generation lentiviral vector. In Figure 1, the authors show both the CMV promoter (arrow) ad a full 5' LTR; furthermore, they use a packaging construct expressing tat. Since this study's final objective is to show the use of IDLV to achieve passive immunotherapy, it would have been more appropriate to remove HIV accessory proteins from the system.
- Likely, the author uses a three plasmid packaging system for IDLV production. Did the authors consider the use of a four plasmid packaging system to improve the safety profile?
- The authors should explain how they established the experimental size used in the in vivo study and add appropriate statistical analysis when possible.
- The authors should show the data obtained in mice challenged at 30 days after the administration of IDLV.
- Did the authors observe background integration using their IDLV? It is essential to show (or provide references to) data supporting the lack of background integration for the system used. In an immunotherapy approach, it is of crucial importance that the production of antibodies is transient and that background integration, which could potentially cause insertional mutagenesis, is not observed.
- Figure 5 panel a, b legends have numerous typos. Furthermore, in panel a, an IDLV-HuIgG is reported and not mentioned in the article body.
Minor concerns
- The authors could consider adding more details regarding the choice of mAb used in this study. For example, other broadly-neutralizing mAbs could have been viewed as more suitable for this study.
- Line 45. The authors use "Eliminating" concerning the risk of IDLV integration to target cells. However, it was shown that background integration could happen with IDLV (and it is usually comparable with the one observed when using plasmid DNA). I think that "drastically reducing" could be more appropriate.
- Line 53. At the sentence "IDLV are pseudotyped" I will add "routinely." VSV-G is not the only protein that could be used.
- Please provide more details in Figure 1
- Line 156. Move ref. [21] to line 154.
Reviewer 2 Report
Michelini et al describe a method for rapid and efficient protection against infectious agents by delivering the genetic information allowing patients cells to produce monoclonal antibodies (mAbs). They use integrase-defective lentiviral vectors to accomplish this and have used the humanised mAb VN04-2, a broadly neutralizing antibody against H5 influenza A virus, as proof of principle.
They show a dose and time-dependent production of the mAb and found that intranasal administration of the vector achieves higher protection compared with the intramuscular route. Importantly, they show that this methodology may be an effective strategy to achieve rapid immunization.
Major points:
1 – The authors show the presence and increment of VN04-2 mAbs in the serum of mice, with the highest amount of IgG achieved by day 28 (Fig.4a). Then on Figure 4c and 4d, there is a little discrepancy on the group of mice that had vectors administered via the i.m. route, where data shows sustained but not increase of the antibody produced. This discrepancy begs the question if the blot used for figure 4a is really a representative animal or is it the animal that achieved highest amount of mAb produced? This should be stated.
More importantly, the authors state that mice challenged 30 days post injection only had partial or no protection, which suggests that mAbs were below the level needed to prevent infection (lines 278-281). But this statement clearly goes against the data shown in Figure 4. So how do the authors explain these observations?
If local concentration is paramount for protection, then this needs be assessed, with a biopsy of the lungs, extraction of proteins and either a western blot or ELISA to measure VN04-2 mAb.
2 – It would be important to understand which cells are producing the mAb in vivo, as well as the differences between expressing cells via the two routes explored in this report. If lentiviral vectors delivered by i.n. route find target cells are more susceptible to infection, then this could explain the observation that this route achieves higher production of mAb, for ex.
3 – The authors indicate that it is possible that the amount of IDLV is reduced over time (line 328). It would then be important to assess if the genetic information provided by the vector is indeed lost. The authors should then access genome vector copies by a quantitative method such as qPCR.
4 – This reviewer thinks that combining rapid protection provided by VN04-2 and a second dose of IDLV expressing NP to confer prolonged protective immunity is an important aspect of this report. Therefore, I suggest moving the supplemental Figure 3 to the main text, as Figure 6.
Minor points:
1 – line 31: Infectious disease -> infectious diseases.
2 – line 44: “non-integration” should be expanded for the general public to understand this feature.
3 – line 67: consider revising the sentence: “Briefly, IDLVs from transfected cell supernatants (…)”
4 – line 107: The sentence “all animal procedure (…)” is a repetition from line 97. Please consolidate and delete as appropriate.
5 – line 152: Please add a reference.
6 – line 157: Please provide a reference for all plasmids used in this study.
7 – line 239: the word “if” is repeated. Please delete one of them.
8 – line 252: Please state the amount of IDLV used in the experiments. And please explain why day 4 was chosen for challenge (line 254).
9 – line 271: The paragraph starting in this line (“to establish the duration of protection (…)”) does not make part of the legend of figure 5. Please start a new paragraph in the main text.
10 – line 317: please amend: “tended to waned” -> tended to wane.
11 – line 327: The sentence “This effect is possibly due to a decrease over time (…) does not make sense taking into account Figure 4. Please see major point 1.
12 – Figure 2: Western blot lanes should provide information regarding the amount of vector used in transduction, either on the top of the gel within the figure or numbering the lanes and providing that information in the legend.
13 – Figure 4: Please consider stating that panel (a) is a representative mouse, if indeed it was (see major point 1).
14 – Figure 5. Please provide full information regarding this experiment in the legend, such as vector quantity, number of mice, etc. Also, please indicate in panels (a) and (d) that mice were sacrificed, [otherwise, it seems that all mice, on average, recovered eventually - not true, taking into consideration panels (b) and (e)], or make a better connection between the two panels. The authors could also consider showing the data for all individual mice, instead of providing the averages, which would show for example on Figure 1a, that the control mice’s weight gain after day 8 (blue line) results from one single survivor. While the other 4 control mice had to be sacrificed before.
Round 2
Reviewer 1 Report
The authors have addressed the concern expressed in the first round of review.
There are some minor format errors that need to be checked during the final article text editing
- line 61: full stop missing
- line 172: H in IDLV-HuG1 is capital, when used in otherpart of the text is lowercase
Reviewer 2 Report
This manuscript is much improved and I believe that the authors answered all the points previously raised. I agree that some points are outside the scope of this paper but it was important to see them discussed in the paper.
There are just a couple of minor points regarding editing:
1) some figure legends are at times separated into another paragraph and seem part of the main text (ex, Figure 2 and Figure 4).
2) lines 306, 336 and 366 - supplemental figure should be number 4